# A Global Scoping Review of the Factors Associated with HIV and Syphilis Co-Infection: Findings from 40 Countries

**Karan Varshney \***, **Alexander Ikanovic, Prerana Ghosh, Pavan Shet, Marcus Di Sipio, Chirag Khatri and Malik Quasir Mahmood**

School of Medicine, Deakin University, Geelong, VIC 3216, Australia; aikanovic@deakin.edu.au (A.I.); ghoshpr@deakin.edu.au (P.G.); pshet@deakin.edu.au (P.S.); mdisipio@deakin.edu.au (M.D.S.); ckha1999@gmail.com (C.K.); malik.mahmood@deakin.edu.au (M.Q.M.)

\* Correspondence: karan.varshney@students.jefferson.edu; Tel.: +61-(03)-5227-1100

**Abstract:** Human immunodeficiency virus (HIV)–syphilis co-infection poses a threat to certain populations, and patients may have considerably poorer health outcomes due to these infections. Our objective was therefore to provide a scoping review of the literature regarding the factors associated with HIV–syphilis coinfection. We searched PubMed, Scopus, and Web of Science, and produced a total of 1412 articles. After completing the screening process as per the Preferred Items for Systematic Review and Meta-Analysis extension for Scoping Review (PRISMA-ScR) guidelines, a total of 109 articles were eligible for inclusion. A total of 68,634 co-infected patients were included in our review. Findings from studies across 40 countries demonstrated that males—particularly men who have sex with men—compose the overwhelming majority of co-infected cases. Additional risk factors include a low CD4 cell count, current or past sexually transmitted infections, and a high number of sexual partners. Our findings have important implications in guiding public health programs across the globe that aim to lower the rates of HIV–syphilis co-infection. More research is also needed on the role of educational attainment, comorbidities, and consistent condom usage regarding the risk for co-infection.

**Keywords:** co-infection; epidemiology; HIV; syphilis; demographics; risk factors





## 1. Introduction

Human immunodeficiency virus (HIV) is a bloodborne retrovirus transmitted through direct contact between blood, broken skin, or mucosal tissue; the virus can be transmitted sexually [1]. The virus targets CD4+ T cells, resulting in gradual immunological loss and abnormalities. This occurs until significant immunodeficiency develops and patients present with oncological or infectious complications, such as Kaposi's sarcoma and toxoplasmosis, associated with a characteristic progression of disease known as acquired immunodeficiency syndrome (AIDS) [2]. In 2020, there were 37,700,000 people living with HIV (PLWH), with 680,000 people dying due to HIV-related causes [3]. While the availability of antiretroviral (ARV) drugs has been valuable for increasing the life expectancy for PLWH across the world, this population nonetheless faces health issues, even if they are on treatment. For example, PLWH tend to be at increased risk for malignancies and cardiovascular disease, and have been shown to report lower health-related quality of life when compared to those who do not have HIV [4–6].

Another important sexually transmitted infection (STI) that has an impact on the health of certain populations is syphilis, which is caused by the spirochetes bacteria *Treponema Pallidum*. It is estimated that, every year, there are approximately 6 million cases of syphilis across the globe amongst individuals aged 15–49 [7]. More than 300,000 annual fetal and neonatal deaths are attributed to syphilis infection [8]. Syphilis spreads through direct contact, with patients developing characteristic syphilitic sores. It follows a typical

progression that can last for years, with symptoms mimicking other diseases [9]. The primary stage of syphilis is characterized by a singular chancre, which is a firm, round, painless lesion, which can progress to a secondary stage characterized by skin rashes and mucous membrane lesions; if left untreated, this may further progress to tertiary syphilis, which can have severe multi-organ consequences; a notable complication is the development of neurosyphilis, which can cause dementia, paralysis, and eventual death [9,10]. While syphilis is curable and preventable, structural barriers such as legal difficulties, policies, criminalization, violence, and discrimination limit its management [11].

While HIV and syphilis pose health concerns for those infected with one of these pathogens, the threat becomes even larger in cases of co-infection. For this reason, the co-infection of the two pathogens is increasingly being understood as a major public health concern. HIV–syphilis co-infection has numerous consequences on health, impacting treatment outcomes, transmission, and the physiological response to infection [12,13]. The downregulation of innate and acquired immune responses in HIV contribute to the increased susceptibility to syphilis co-infection [14]; reciprocally, syphilis has also been shown to increase HIV transmission through increased viral loads and decreased CD4+ T-cells in PLWH receiving ARV treatment [14]. Along with the lower CD4 cell count being an important immunological marker of HIV progression, the CD4/CD8 ratio has been shown to be an important marker associated with non-AIDS diseases, such as diabetes, pulmonary emphysema, and coronary artery disease; notably, a low CD4/CD8 ratio has been associated with higher morbidity and mortality [15–17]. Critically, HIV–syphilis co-infection has been shown to worsen immune recovery, decrease ARV effectiveness, and increase the risk of neurocognitive and ophthalmic issues [12,18].

In consideration of the major health risks posed by HIV and syphilis co-infection, there is a clear need to determine the factors relating to co-infection, and to understand what populations are at the highest risk. Having a clear understanding of such risks will assist in public health programs that can maximize prevention and ensure the best possible health outcomes for vulnerable populations. Therefore, the purpose of our work was to conduct a scoping review of the literature on risk factors associated with HIV–syphilis co-infection, particularly amongst vulnerable populations.

## 2. Materials and Methods

Our scoping review workflow followed the "Preferred Items for Systematic Review and Meta-Analysis extension for Scoping Review" (PRISMA-ScR) guidelines [19,20]. The searches were conducted on 16 February 2022, in three different databases: Scopus, PubMed, and Web of Science. There were no restrictions placed based on date of publication. Our searches included terminology on HIV, syphilis, and co-infection.

Two reviewers (KV and AI) independently screened the articles after retrieval from the three databases. After the removal of duplicates, the papers were screened based on title/abstract, and full text thereafter. The articles were eligible for inclusion if they met the following criteria:

- they were written in English,
- they were quantitative, original research,
- they included at least five patients co-infected with HIV and syphilis,
- they included data on demographics/factors associated with infection,
- they provided stratified data for HIV–syphilis co-infected patients.

The data collection involved the extraction of data pertaining to the study characteristics and patient characteristics, respectively. The data extracted relating to study characteristics were the year of publication, the country where the study took place, the study design, and the total number of co-infected individuals. The patient characteristics included the demographics, health status, and additional factors. The demographic data extracted included gender (Male, Female, Transgender), age (<25, 25–34, 35–44, >45), sexual orientation (MSM/Homosexual, Bisexual, Heterosexual), and race/ethnicity (Black/African, Hispanic/Latino, Asian, Multi-race, Other). While it is important to acknowledge that

MSM and homosexual are distinct categories of sexual identity, we classified them as a single category in our review because a high number of included studies did not separately classify MSM and homosexual men, and instead classified them together.

The health status data included other infections/morbidities, whether they had a prior history of syphilis/reinfection, the CD4 cell count, the HIV viral load, and whether they were on ARVs. The additional factors included any other data that may have served as risk factors; these included: number of sexual partners, condom usage, martial/relationship status, substance use, educational attainment, and history of other STIs. After the collection of the data, the data were pooled and placed in frequency tables. Additional stratification was completed and pooled for high-risk populations.

## 3. Results

### 3.1. Searches and Articles Included

Searches from the databases produced a total of 1412 articles. After the removal of duplicates, 1256 articles remained. Of these articles, 237 remained after screening by title/abstract. Next, after screening by full text based on our inclusion criteria, a total of 109 articles were deemed to be eligible for our review [21–129]. Articles were excluded during the stage of full-text analysis for the following reasons: they were not in English, they did not describe risk factors for co-infection, they did not stratify for those who are co-infected, they had fewer than five co-infected patients, they were not original research, and they did not include any HIV–syphilis co-infected patients. Figure 1 outlines our workflow process.

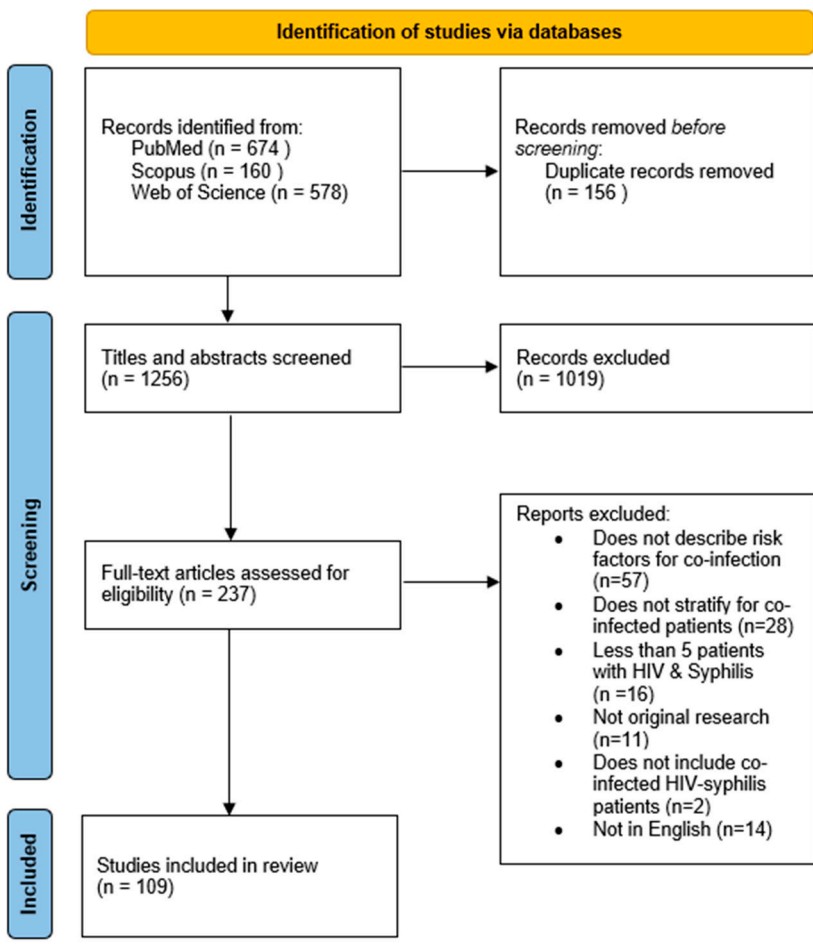

**Figure 1.** Process of searching and screening articles (figure adapted from Page et al., 2021 [17]).

## 3.2. Study Characteristics

The 109 included studies were conducted in 40 different countries. The study totals across continents were the following: Africa (n = 15), Asia (n = 32), Australia (n = 1), Europe (n = 22), North America (n = 32), and South America (n =12). The highest numbers of studies were conducted in the following countries: the United States of America (USA) (n = 25), China (n = 18), Brazil (n = 10), Canada (n = 5), and Taiwan (n = 5). The most frequently utilized study design was a retrospective cohort study (n = 62), followed by cross-sectional analysis (n = 26), a prospective cohort study (n = 14), a case-control study (n = 4), and lastly, a case series (n = 3). The majority of the articles were published after 2010. Table 1 summarizes the study characteristics.

**Table 1.** Characteristics of the included studies.

| Characteristic | Count (%) |
| --- | --- |
| *Total studies* | 109 (100) |
| *Total co-infected patients (range)* | 68,634 (8–21,183) |
| *Total countries* | 40 (100) |
| *Median number of studies by country (range)* | 1 (1–25) |
| *Study design* | |
| Retrospective cohort study | 62 (56.88) |
| Prospective cohort study | 14 (12.84) |
| Cross-sectional design | 26 (23.85) |
| Case-control study | 4 (3.67) |
| Case series | 3 (2.75) |
| *Year of Publication* | |
| 2017–2022 | 57 (52.23) |
| 2011–2016 | 40 (36.70) |
| 2003–2010 | 12 (11.01) |

## 3.3. Patient Characteristics

There was found to be a pooled total of 68,634 co-infected patients. The pooled patient characteristics are shown in Table 2.

**Table 2.** Pooled analysis of factors associated with syphilis–HIV co-infection.

| Factor | Count (%) |
| --- | --- |
| *Total co-infected patients (n = 68,634)* | |
| *Gender (n = 59,420)* | |
| Male | 53,725 (90.42) |
| Female | 5550 (9.34) |
| Transgender | 145 (0.24) |
| *Sexual orientation (n = 41,524)* | |
| MSM/homosexual | 36,138 (87.03) |
| Bisexual | 221 (0.53) |
| Heterosexual | 5165 (12.44) |
| *Age (n = 22,819)* | |
| <25 | 5462 (23.94) |
| 25–34 | 8542 (37.43) |
| 35–44 | 4165 (18.25) |
| >45 | 4650 (20.38) |

**Table 2.** *Cont.*

| Factor | Count (%) |
|---|---|
| *Race/Ethnicity (n = 47,143)* | |
| White/Caucasian | 18,040 (38.27) |
| Black/African | 16,491 (34.98) |
| Hispanic/Latino | 9111 (19.33) |
| Asian | 1678 (3.56) |
| Multi-race | 472 (1.00) |
| Other | 1351 (2.87) |
| *Infections (n = 3158)* | |
| Concomitant STIs * | 3158 (100) |
| Hepatitis B/C | 1229 (38.92) |
| Human papillomavirus (HPV) | 672 (21.28) |
| Herpes Simplex Virus (HSV) | 351 (11.11) |
| Gonorrhea | 309 (9.78) |
| Chlamydia | 72 (2.27) |
| *Reinfection/prior history of syphilis (n = 7682)* | |
| Yes | 2951 (38.41) |
| No | 4731 (61.59) |
| *Comorbidities (n = 1875)* | |
| Psychiatric diagnoses | 1875 (100) |
| *CD4 cell count (n = 6045)* | |
| <200 | 1680 (27.79) |
| >200 | 4365 (72.21) |
| <350 | 2290 (37.88) |
| >350 | 2283 (37.77) |
| <500 | 3964 (65.57) |
| >500 | 1123 (18.58) |
| *Viral load (n = 5553)* | |
| Detectable/unsuppressed/<200 copies/mL | 2408 (43.36) |
| Undetectable/suppressed/>200 copies/mL | 3145 (56.64) |
| *On ARVs (n = 5011)* | |
| Yes | 3888 (77.59) |
| No | 1123 (22.41) |
| *More than 1 sexual partner (n = 1995)* | |
| Yes | 1533 (76.84) |
| No | 462 (23.16) |
| *Condom usage (n = 861)* | |
| Yes | 427 (49.59) |
| No | 434 (50.41) |
| Always/consistent | 244 (28.34) |
| Sometimes/inconsistent | 73 (8.48) |
| *Marital/relationship status (n = 2935)* | |
| Married/cohabiting/polygamy | 917 (31.24) |
| Unmarried/single | 1610 (54.86) |
| Divorced/widowed | 408 (13.90) |
| *Injection drug use (n = 1163)* | |
| Yes | 645 (55.46) |
| No | 518 (44.54) |
| *Education (n = 4354)* | |
| Less than secondary school/ incomplete secondary school | 280 (6.43) |
| Completed secondary school | 1297 (29.79) |
| College/university/trade school | 2777 (63.78) |

* Referring to the presence of any additional STIs, including (but not limited to) HPV, HSV, gonorrhea, chlamydia, and trichomoniasis. Note: a number of studies did not specify the concomitant STI.

### 3.3.1. Patient Demographics

The co-infected patients were shown to be overwhelmingly male, with a total of 53,725 males (90.42%). A small proportion of patients (0.24%) were transgender. Riley et al. (2020) demonstrated that males had 2.43-times higher odds of co-infection compared to females (95% CI: 1.90–3.12) [93].

In terms of sexual orientation, the co-infected patients were largely either MSM or homosexual (87.03%), with heterosexual patients constituting only 12.44% of the co-infection cases. Tullio et al. (2021) showed that MSM contact was associated with a 2.64-times higher odds of co-infection compared to those who did not have such contact/had heterosexual contact only (95% CI: 1.48–4.72) [117]. Similar findings were found in a study by Thurnheer et al. (2010), where MSM have 3.39-times higher odds of co-infection compared to heterosexuals (95% CI: 2.70–4.26) [112].

Patients across all age groups were shown to be at risk for co-infection. Nonetheless, adults aged 25–34 were the age group who were most frequently co-infected (37.43%), followed by those under 25 (23.94%). While there were no clear patterns regarding race/ethnicity, it was shown that White/Caucasian and African/Black individuals were at the highest likelihood of being co-infected.

### 3.3.2. Health Status

The infection of microbes (in addition to HIV and syphilis) was noted in a proportion of cases, though studies which accounted for additional infections focused primarily on additional STIs. The most frequent STI poly-infections were Hepatitis B/C (n = 1229), human papillomavirus (n = 672), and herpes simplex virus (n = 351). A prior history of syphilis was also noted for a number of patients (n = 2951). Across the 109 studies, only one study, which was conducted in South Korea (Lee et al., 2020), focused on comorbidities; it was found that 1875 (out of a total of 4536 patients) had some form of psychiatric diagnosis [68].

There did not appear to be major trends for co-infection in terms of whether one had a detectable HIV viral load or not. Similarly, being on ARV did not decrease the frequency of co-infection. Those with a CD4 cell count of <200 and <350 had frequencies of co-infection less than and comparable to those with >200, and >350 CD4 counts, respectively. However, there were considerably more individuals with a CD4 count <500 (n = 3964) compared to those with a CD4 count >500 (n = 1123).

### 3.3.3. Additional Factors

Of the additional factors, having more than one sexual partner (76.84%), being an injection drug user (55.46%), being unmarried/single (54.86%), and having a post-secondary education (63.78%) were the most frequent among those with co-infection. There were not notable differences in co-infection between those who used condoms (n = 427) and those who did not use condoms (n = 434).

### 3.4. Factors Amongst MSM

In total, 17 of the 109 included studies provided an analysis that was restricted to MSM, which encompassed 1605 patients. Table 3 lists the details of the pooled analysis for this group.

There were no notable trends by age. By race/ethnicity, Asians made up a sizeable proportion of the cases (31.75%), which was more than any other racial/ethnic category. While additional infections were not frequent among co-infected MSM patients, it is worth noting that the most common infections were HPV (n = 100) and hepatitis (n = 93). A past history of STIs (which included syphilis) was present in 151 cases. As was the case with the other patients of the pooled analysis, a high proportion of co-infected MSM patients were on ARVs (95.76%) and had an undetectable HIV viral load (81.09%). There were more patients with a CD4 count <500 than patients with a CD4 count >500. Similarly to the findings for all of the co-infected patients, the rates of co-infection for MSM were highest

among those who were unmarried/single (70.12%), had post-secondary education (41.02%), used condoms (61.65%), and had more than one sexual partner (84.10%). Uniquely to the MSM patients, alcohol use (n = 96) and having receptive anal sex (n = 142) were notable risk factors.

**Table 3.** Pooled analysis of factors associated with syphilis–HIV co-infection amongst studies restricted to MSM (total studies = 17).

| Factors | Count (%) |
|---|---|
| *Total MSM (n = 1605)* | |
| *Age (n = 658)* | |
| <25 | 37 (5.62) |
| 25–34 | 188 (28.57) |
| 35–44 | 216 (32.83) |
| >45 | 217 (32.98) |
| *Race/Ethnicity (n = 756)* | |
| White/Caucasian | 205 (27.12) |
| Black/African | 156 (20.63) |
| Hispanic/Latino | 72 (9.66) |
| Asian | 240 (31.75) |
| Other | 83 (10.98) |
| *Infections (n = 260)* | |
| Hepatitis B/C | 93 (35.77) |
| Herpes Simplex Virus (HSV) | 67 (25.77) |
| Human papillomavirus (HPV) | 100 (38.46) |
| *History of STI (including syphilis) (n = 151)* | |
| *CD4 count (n = 275)* | |
| >500 | 115 (41.82) |
| <500 | 160 (58.18) |
| *Viral load (n = 476)* | |
| Detectable/unsuppressed (>200 copies/mL) | 91 (19.12) |
| Undetectable/suppressed (<200 copies/mL) | 386 (81.09) |
| *On ARVs (n = 118)* | |
| Yes | 113 (95.76) |
| No | 5 (4.24) |
| *Marital/relationship status (n = 251)* | |
| Married/cohabiting/polygamy | 61 (24.30) |
| Unmarried/single | 176 (70.12) |
| Divorced/widowed | 14 (5.58) |
| *Education (n = 529)* | |
| Less than secondary school/incomplete secondary school | 111 (20.98) |
| Completed secondary school | 201 (38.00) |
| College/university/trade school | 217 (41.02) |
| *Condom usage (n = 339)* | |
| Yes | 209 (61.65) |
| No | 130 (38.35) |
| Always/consistent | 32 (9.44) |
| Sometimes/inconsistent | 64 (18.88) |
| *Substance use (n = 244)* | |
| Any use of substances | 244 (100) |
| Illicit substance use | 69 (28.28) |
| Tobacco use | 6 (2.46) |
| Alcohol use | 96 (39.34) |

**Table 3.** *Cont.*

| Factors | Count (%) |
|---|---|
| *More than 1 sexual partner/casual sex (n = 415)* | |
| Yes | 349 (84.10) |
| No | 66 (15.90) |
| *Role in anal sexual intercourse (n = 158)* | |
| Insertive | 16 (10.12) |
| Receptive | 142 (89.87) |
| Receptive and Insertive | 38 (24.05) |
| Receptive only | 104 (65.83) |

## 4. Discussion

While the risk factors for HIV mono-infection and syphilis mono-infection have been described extensively in the past, our findings provide major contributions by offering concrete insights regarding vulnerabilities for concurrent infections of both of these pathogens. Considering that co-infection patients have been shown to have worse outcomes compared to those who are only infected with one of these two pathogens, our findings have important public health implications, and can offer insights into future programs as to how interventions can be tailored to most effectively reduce morbidity and mortality.

Our findings highlight that those who are most vulnerable to HIV–syphilis co-infection are male and MSM. These are findings that are consistent and hold in contexts across the globe. Our findings contribute to the literature in a number of important ways. Critically, they demonstrate that there is a clear need to recognize the vulnerability of MSM/homosexual men to co-infection. Among this demographic, those with a high number of sexual partners, those who are aged 25–34, and those who inconsistently use condoms are at the most vulnerable. Our findings also demonstrate that, despite the widespread availability of ARVs across the globe, a high proportion of individuals were shown to have high viral loads and low CD4 cell counts, which is likely to have contributed to the contraction of syphilis. There is therefore a clear need to improve access to ARVs and to increase adherence overall. Considering that the COVID-19 pandemic has negatively impacted ARV adherence across different settings [130–132], there is an urgent need for the enactment of policy that can remove barriers to accessing HIV care, and to lower the financial costs of treatment. This can happen at the national level for countries by increasing the funding of public health budgets towards the provision of resources for HIV care, providing patient-centered delivery schedules of ARV prescription refills, and utilizing electronic dispensing tools for medications and adherence monitoring [133–135].

MSM are the most vulnerable group to HIV–syphilis co-infection. Therefore, public health programs need to be developed and focused on supporting MSM who are currently living with these infections. Interventions focusing on prevention should hence also be directed towards MSM with mono-infection of HIV or syphilis, while such interventions concurrently raise awareness of the dangers of co-infection. One such program is currently being developed, where an app equips MSM with the tools to better understand their risk of contracting an STI, thus helping them become more aware of the level of their risk when having sex with new partners [136]. Having MSM sexual health influencers encourage peers to test for HIV and/or syphilis has shown potential in encouraging MSM to self-test, and may also offer utility in a public health context [137]. Policy considerations to further support MSM can revolve around destigmatizing efforts, both for co-infection and for MSM as a whole. Considering that HIV stigma, particularly towards MSM, in the healthcare setting has been described as a deterrent to seeking care [138,139], policy considerations should be made for the removal of such barriers for these vulnerable individuals.

Among MSM, having receptive anal sex has also been shown to be an important vulnerability for co-infection. This is a pattern that has also been shown to hold for STI transmission in general, with a proposed explanation being that an insufficient amount of

lubrication of the rectum area can lead to mucosal trauma, and therefore higher vulnerability to STI transmission [140]. While condom usage was not shown to be protective in our review, it is important to note that there were very limited data on the consistency of condom usage among MSM (and other populations). Therefore, it is imperative that future research determines the extent to which condoms are being used on a consistent basis, while programs are concurrently being implemented to increase the rates of condom usage among MSM. Utilizing methods that are culturally and racially specific may be effective in the promotion of the usage of condoms in order to lower STI transmission rates [141].

For MSM, and for all other populations, there were a number of similar findings regarding co-infection vulnerability. While a large proportion of patients were on ARVs and had a relatively low viral load, a high proportion of patients also had relatively low CD4 counts; this may have been an important risk factor that should not be neglected, as it indicates that individuals may be on ARVs but adhering to treatments at rates that are lower than optimal. There is thus a need for the upscaling of HIV treatment adherence programs, especially considering the many ways in which the COVID-19 pandemic has contributed to difficulties in maintaining ARV adherence [142,143].

There were shown to be some differences in race/ethnicity. Amongst the general population, White/Caucasian individuals were found to have the highest rates of co-infection, followed by Black/African populations. Amongst MSM, the highest rates of co-infection were amongst Asian populations. It is important to note that, in a number of nations, Asian and Black/African populations make up only a small proportion of the population. Possible reasons for the disproportionately high rates in Asian and Black/African groups may include a lack of culturally competent care, and racial prejudice/discrimination in healthcare settings. This further emphasizes a need for care that offers more cultural sensitivity, and to remove unjust barriers in healthcare settings.

Concomitant or prior history of STIs was shown to be a risk factor for infection, particularly hepatitis B/C, HPV, and previous syphilis infections. However, there were little data on the role of other infections in possibly contributing to HIV–syphilis co-infection risk. Furthermore, despite chlamydia and gonorrhea being some of the most common STIs [144], few HIV–syphilis co-infected patients were shown to have infections of these microbes; there is therefore a clear need to better understand the prevalence of these pathogens among co-infected patients, and to understand the unique factors that increase the risk for contraction of these additional STIs. Additionally, there were no data on the role of physical illnesses with regard to risk. There is therefore a need for future research to focus on such factors, as well as the possible role of mental health comorbidities in increasing overall vulnerability. It is worthwhile to also note that educational attainment paradoxically increased the risk for co-infection. The reasons for this remain unclear, and should be investigated further in future research.

Alongside our findings, it is important to consider the limitations of our work. First of all, little can be said in terms of the temporality of the findings. It is unclear if certain factors increased the risk for co-infection, or if co-infection influenced certain risk factors and/or parameters for patients. Connected to this point, we were unable to clearly distinguish whether patients first contracted syphilis or HIV, or if the infections occurred simultaneously. Additionally, while our findings provided an understanding of an array of factors associated with co-infection, our review offers limited insights into the lived experiences of patients, and the decisions made that increased or decreased the risk for certain behaviors that may have led to infection. Regardless of such limitations, it is worthwhile to note that the large number of studies included in this review across cultural contexts indicates the high level of robustness of our findings, and the applicability of the work in shaping public health programs and policy.

## 5. Conclusions

Our work highlighted a number of important factors that increase the vulnerability of HIV–syphilis co-infection. These factors include being male, being MSM, having a low

CD4 cell count, having a high number of sexual partners, being single, and having prior or current STIs. Our findings can be applied to improve the overall health outcomes for vulnerable and marginalized groups by informing programs and policies.

**Author Contributions:** Conceptualization, K.V., A.I., P.G., M.Q.M.; methodology, K.V., A.I.; analysis, K.V., P.G.; data curation, K.V., A.I., P.G., P.S., C.K., M.D.S.; writing—original draft preparation, K.V., A.I., P.G., M.D.S.; writing—review and editing, K.V., A.I., P.G., P.S., C.K., M.D.S., M.Q.M.; supervision, M.Q.M. All authors have read and agreed to the published version of the manuscript.

**Funding:** This research received no external funding.

**Institutional Review Board Statement:** Not applicable.

**Informed Consent Statement:** Not applicable.

**Data Availability Statement:** Not applicable.

**Conflicts of Interest:** The authors declare no conflict of interest.

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
