# Peer review of "A Global Scoping Review of the Factors Associated with HIV and Syphilis Co-Infection: Findings from 40 Countries"

_venereology, doi:10.3390/venereology1010007_

Round 1
Reviewer 1 Report
The review is well prepared; a lot of work was done.
Some remarks are below:
Introduction- the sentence should be changed: is: ......CD4/CD8 ratio is a marker of non-AIDS diseases [15]. The authors should find another references indicating co-morbidities other than AIDS. It is simplification. We are talking a lot about non-AIDS diseases, which are the main problem of HIV infected pts, but can not simply indicate such a marker.
Ihe other finding is the low number and ratio of chlamydial infections, even lower than HPV and HSV. The author comment is lacking. The reviewer (as a practitioner) can not believe that it is probable. A wider comment of these results is required.
Reviewer 2 Report
The authors conducted a scoping review of factors associated with HIV and syphilis co-infection. Overall the paper is interesting and well done. I recommend the following improvements to improve the language and clarity of the paper.
- When describing categorical demographics, I recommend including some examples of possible values in parentheses, such as the following: ...race (Asian, Black, White).
- I'd recommend "substance use' rather than "whether they were a drug user", as the former is less stigmatizing.
- I'd probably delete the statement that the study protocol wasn't registered (unless the journal requires it).
- For Table 2, I'd recommend separating MSM from homosexual, as many MSM are bisexual. MSM should be its own category here.
- A footnote should be added describing what "Concomitant STIs" refers to, considering specific STIs are already listed in the table.
- The first paragraph is written from the point that being male and MSM are the key risk factors; I would strongly recommend describing males and MSM as the most vulnerable to HIV-syphilis co-infection (in general, it's good destigmatizing practice to not describe identities as risk factors).
- Some discussion on racial/ethnic differences in HIV/syphilis infection (and potential drivers of these disparities) isa warranted. Especially considering the racial demographic breakdown (which is quite different from the general population).
- More discussion on policy recommendations would be helpful.
- Overall, a much stronger case needs to be made in the discussion describing how this adds to the literature.
Round 2
Reviewer 2 Report
The authors thoughtfully addressed my concerns.